# 3D genome organization contributes to genome instability at fragile sites

Dan Sarni [1], Takayo Sasaki [2], Michal Irony Tur-Sinai [1], Karin Miron[1], Juan Carlos Rivera-Mulia [2,7], Brian Magnuson[3,4], Mats Ljungman[4,5,6], David M. Gilbert [2✉] & Batsheva Kerem [1✉]

Common fragile sites (CFSs) are regions susceptible to replication stress and are hotspots for chromosomal instability in cancer. Several features were suggested to underlie CFS instability, however, these features are prevalent across the genome. Therefore, the molecular mechanisms underlying CFS instability remain unclear. Here, we explore the transcriptional profile and DNA replication timing (RT) under mild replication stress in the context of the 3D genome organization. The results reveal a fragility signature, comprised of a TAD boundary overlapping a highly transcribed large gene with APH-induced RT-delay. This signature enables precise mapping of core fragility regions in known CFSs and identification of novel fragile sites. CFS stability may be compromised by incomplete DNA replication and repair in TAD boundaries core fragility regions leading to genomic instability. The identified fragility signature will allow for a more comprehensive mapping of CFSs and pave the way for investigating mechanisms promoting genomic instability in cancer.

[1] Department of Genetics, The Life Sciences Institute, Hebrew University, Jerusalem 9190401, Israel. [2] Department of Biological Science, Florida State University, Tallahassee, FL 32306, USA. [3] Department of Biostatistics, School of Public Health, University of Michigan, Ann Arbor, MI 48109, USA. [4] Rogel Cancer Center and Center for RNA Biomedicine, University of Michigan, Ann Arbor, MI 48109, USA. [5] Department of Radiation Oncology, University of Michigan Medical School, Ann Arbor, MI 48109, USA. [6] Department of Environmental Health Sciences, School of Public Health, University of Michigan, Ann Arbor, MI 48109, USA. [7] Present address: Department of Biochemistry, Molecular Biology and Biophysics, University of Minnesota Medical School, Minneapolis, MN 55455, USA. ✉email: gilbert@bio.fsu.edu; batshevak@savion.huji.ac.il

DNA replication occurs in a temporal order known as the replication timing (RT) program, which is tightly regulated to ensure the faithful duplication of the genome[1]. DNA replication stress leads to formation of DNA damage and subsequently to genomic instability[2–5]. Replication stress induced by DNA polymerase inhibitors or activation of oncogenes leads to decreased replication fork progression, stalled forks, and activation of dormant origins[4,6–8]. However, dormant origins activation is not always sufficient and the perturbed replication induces chromosomal instability at specific regions termed common fragile sites (CFSs)[9]. CFSs manifest as breaks, gaps, and constrictions on metaphase chromosomes under mild replication stress[10]. CFS instability results from mitotic entry before the completion of DNA replication[11], resulting in DNA breaks and DNA-damage response activation. Failure to complete replication and repair of DNA damage at CFSs results in the transmission of damage and genomic instability across cell generations[12]. CFSs are preferentially unstable in pre-cancerous lesions and during cancer development thus playing a role in driving cancer development[13–16]. In fact, several fragile sites were found within tumor suppressor genes such as *FHIT* and *WWOX* that are unstable in a wide variety of cancer types[17,18].

Several features characterizing CFSs have been proposed to contribute to CFS expression (fragility), including late replication[19,20] and a paucity of replication origins[20,21], which under replication stress impede the completion of their replication. DNA secondary structures induced by AT-rich sequences have also been suggested to cause CFS expression by raising potential barriers to replicating forks[22,23]. In addition, CFSs are enriched with large genes[24] that may contribute to fragility due to replication–transcription collisions[25–27]. However, none of the suggested factors on their own is sufficient to induce fragility since they are prevalent across the genome including non-fragile regions. Furthermore, the suggested fragile site features could not account for all known CFSs[28]. Thus, the molecular basis underlying recurrent chromosomal instability remains unknown. CFS instability was found to be both cell-type- and stress inducer-specific, raising the possibility that fragility is driven by perturbed regulation of organized yet dynamic cellular programs, such as DNA replication, transcription and genome organization. The DNA RT program is temporally and spatially organized[29]. The three-dimensional (3D) organization of the genome, determined by high-resolution chromosome conformation capture (Hi-C), identified domains of chromatin interactions defined as topologically associated domains (TADs)[30,31]. TADs are functional genomic units playing a role in transcriptional regulation[32,33]. Interestingly, TADs associate with replication domains, RT units varying in size between 400 and 800 kb, that change during cell differentiation[34]. Moreover, chromosome architecture was suggested to play a role in maintenance of genome stability[35–38], controlling damage repair by regulating spreading of DNA-damage response signals, such as γH2AX within TADs, with a prominent tendency to stop at TAD boundaries[36,39]. Importantly, the potential role of chromosome architecture in regulating CFS instability has not yet been investigated. Therefore, we hypothesized that a combination of several factors renders a region sensitive to replication stress and induces fragility.

Here we explore the effect of mild replication stress induced by the DNA polymerase inhibitor aphidicolin (APH) on genome-wide replication and transcription programs, using Repli-seq and Bru-seq, and integrate these data with Hi-C data from the 4DN consortium. The results reveal a multilayer chromosomal fragility signature upon replication stress, comprised of a combination of a TAD boundary overlapping an actively transcribed large gene with APH-induced RT delay. We show that the temporal order of replication of only a small part of the genome is altered by APH, but the delayed portion is highly enriched for CFSs. The induced RT delay at CFSs generates a V-shaped RT profile spanning TAD boundaries, within transcribed large genes. Moreover, we show that the CFS core fragility regions (CFR) in unperturbed cells are replicating in mid-S phase and thus are not merely the latest replicating regions as previously suggested[19–21]. Our fragility signature enabled greater precision of mapping the CFR and identification of novel fragile sites that were not detected cytogenetically, highlighting the improved sensitivity of our approach for identifying fragile sites. Altogether, the results suggest that replication, transcription, and genome organization combined underlie CFS expression.

## Results

**Aphidicolin affects the RT of specific genomic regions**. To study the effect of replication perturbation on the replication program we performed Repli-seq on immortalized human fibroblasts (BJ-hTERT). Unchallenged cells were compared with cells treated with a low concentration (0.2 μM) of APH for 24 h, commonly used to induce CFS expression (Fig. 1a), thus cells were exposed to APH for at least one cell cycle. Such mild replication stress had no effect on the progression of cells through S phase as indicated by analysis of pulse-chase-pulse labeling cells with thymidine analogs (Supplementary Fig. 1). Genome-wide correlation analysis of RT profiles was able to distinguish control from APH-treated cells (Fig. 1b), despite a strong genome-wide correlation among all samples (Fig. 1b). Likewise, principal component analysis showed clustering of RT profiles according to the treatment (Fig. 1c), further supporting that APH affected the RT. Next, we extracted all RT-variable regions by segmenting the genome into 100 kb windows and applied an unsupervised K-means clustering analysis to genomic regions that changed their RT by at least 1 RT unit (two standard deviations in pairwise comparisons of technical repeats) (Supplementary Fig. 2a), as previously described[40]. We identified 1277 RT-variable regions (4% of the genome), consistent with the correlation matrix analysis (Fig. 1b). Many RT-variable windows were contiguous and, when assembled together, identified 607 differentially replicated regions. Clustering the RT-variable regions into five clusters identified specific RT signatures containing early replicating regions that are delayed by APH (RT signature 3) and late replicating regions that were advanced in APH-treated cells (RT signature 1) (Fig. 1d, e, Supplementary Fig. 2b). Moreover, we identified regions where RT changed within the same fraction of S phase (early or late, RT signatures 2, 4, and 5) (Fig. 1d, e, Supplementary Fig. 2b). For visualization of RT profiles and downstream analyses RT data were binned into 5 kb nonoverlapping windows. It is worth noting that 2/3 of RT-variable regions (387) were delayed by APH (Supplementary Data 1). These loci are of interest for further investigation of CFS instability as mild replication stress induces under-replication associated with CFS expression[11,19].

To test whether CFSs are enriched with RT-variable regions we analyzed the RT of the CFS bands we have previously identified in BJ-hTERT cells[41]. The analysis showed APH-induced RT-delayed loci in 20/24 (83%) of identified CFS bands (Fig. 1f, Supplementary Fig. 3). Advanced RT regions were found only in 8/24 (33%) CFS bands, amongst them 7 CFS bands harbored both advanced and delayed RT loci (Fig. 1f, Supplementary Fig. 3). These results indicate that CFSs are enriched for delayed RT. Next, we trisected the RT program into early, mid and late S phase according to the RT value. We compared the RT of APH-induced delayed regions in control cells and revealed that 85% of the RT-delayed loci in CFSs are

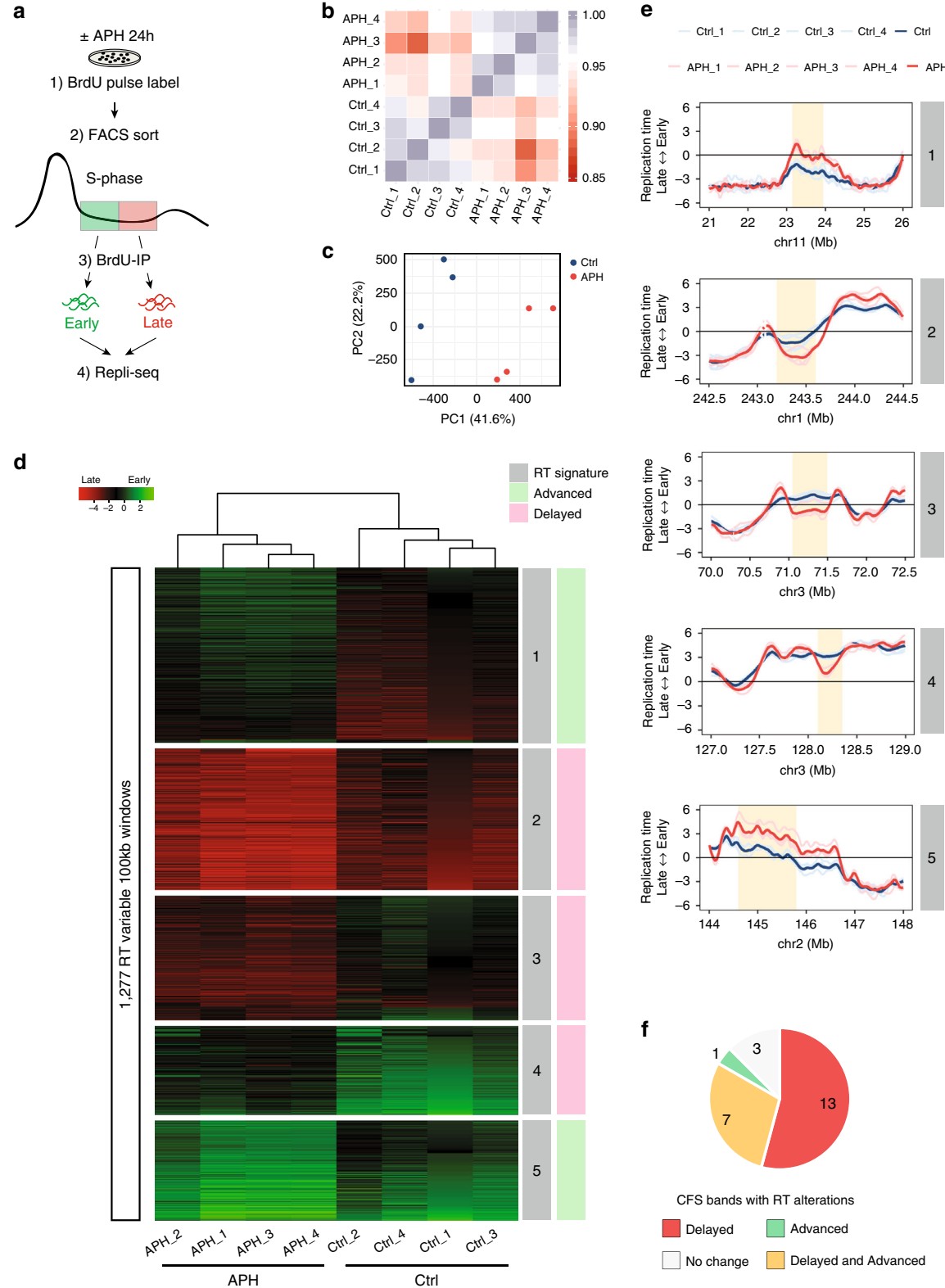

replicated in early/mid-S phase (Supplementary Fig. 4). Thus, RT-delayed loci are not the latest replicating regions in unchallenged cells.

**Delayed replication of large genes at CFSs is associated with fragility**. Cytogenetic mapping enables the identification of fragile sites on metaphase chromosomes of cells grown under replication stress. However, due to its low resolution, cytogenetic mapping does not allow the identification of the CFR within a band harboring a CFS. In order to investigate whether the delayed replication is the CFR within CFSs we used molecular mapping using BAC clones as fluorescence in situ hybridization (FISH) probes on metaphase chromosomes from cells treated with APH. We first focused on two CFSs, at 2q22 and 22q12, that we have previously identified in BJ-hTERT cells[41].

**Fig. 1 Aphidicolin affects the RT of specific genomic regions. a** Schematic description of Repli-seq. BJ-hTERT cells with (+) or without (−) APH treatment were pulse labeled with BrdU and sorted into early and late S phase, and the RT programs were obtained by next-generation sequencing. **b** Correlation matrix of genome-wide RT programs of BJ-hTERT cells with (APH) or without (Ctrl) APH treatment. Four replicates per condition are presented (1–4). **c** Principal component analysis (PCA) of genome-wide RT showing the 1st and 2nd components for control (blue) and APH-treated cells (red). **d** Identification of APH-specific RT signatures, distinguishing APH-treated cells (APH) from non-treated control cells (Ctrl). Unsupervised k-means clustering analysis of RT-variable regions identified specific RT signatures (labeled in gray boxes). The heat map shows the RT ratios [= log2(early/late)]. **e** Exemplary RT profiles of RT signatures. Numbering according to RT signature in (**d**). The RT profiles are displayed as log2 ratios of signals from early and late S-phase fractions. Positive values correspond to early replication, and negative values correspond to late replication. Four replicates per condition are presented color coded according to the legend (Ctrl in blue, APH in red). Yellow boxes mark the RT-variable regions. **f** Pie chart indicating RT alterations within CFS cytogenetic bands mapped in BJ-hTERT cells following APH treatment.

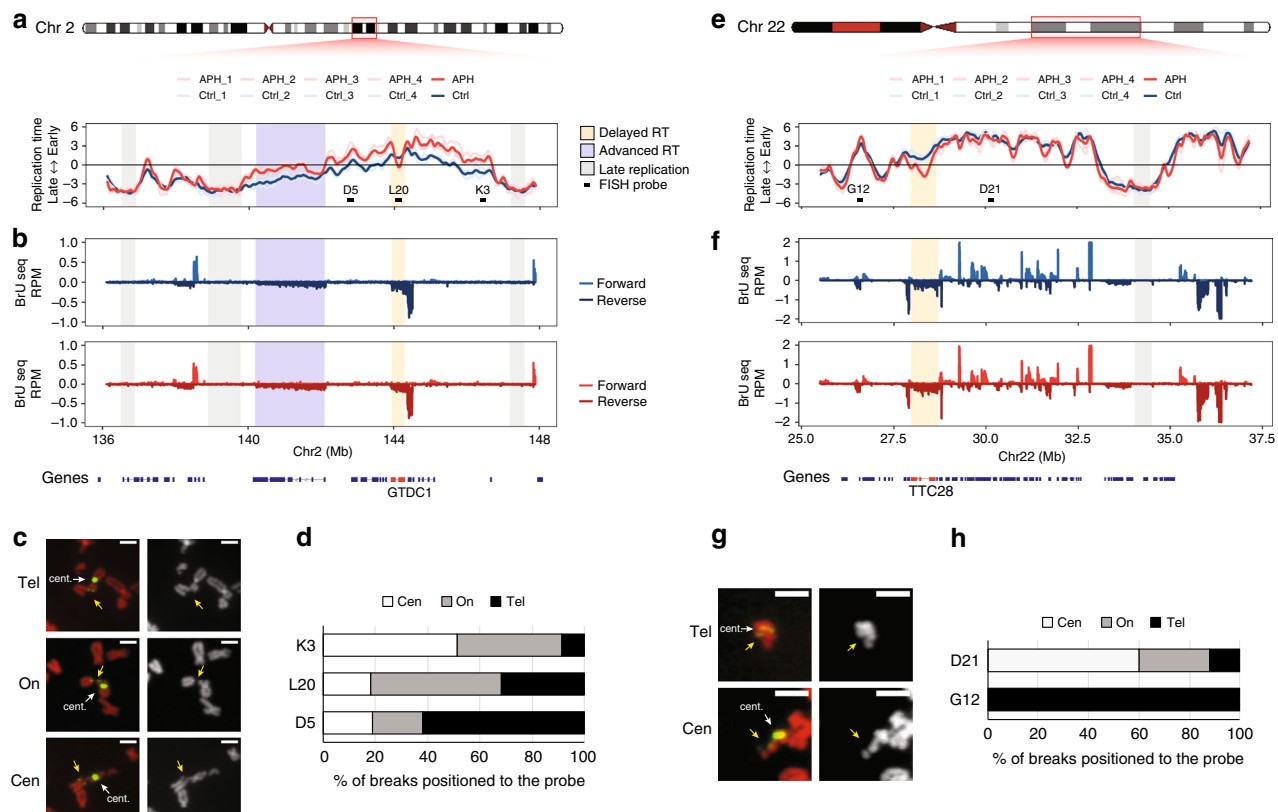

**Fig. 2 Delayed replication at CFSs is associated with fragility. a** RT profile of the fragile site in 2q22. Gray boxes: the latest replicating regions within the 2q22; blue box: the non-fragile LRP1 gene; yellow box: fragility core within GTDC1; black boxes: FISH probes. Four replicates per condition are presented and color coded (Ctrl—blue, APH—red). **b** Nascent RNA transcription, Bru-seq, traces for Ctrl and APH-treated cells at 2q22. Positive values: forward strand, negative values: reverse strand. One replicate per condition is presented and color coded as in (**a**). Annotated genes are shown at the bottom. **c** Representative images of chromosomal breaks on metaphase spreads from BJ-hTERT cells treated with APH and labeled with a FISH probe, RP11-265L20. Left panel: propidium iodide (PI) staining of chromosomes—red; RP11-265L20 FISH and centromere 2 (cent.) probes—green. Right panel: PI contrast staining, yellow arrows—breaks/gaps at the RP11-265L20 probe; white arrows—centromere 2 probe. Breaks position relative to the probe: centromeric (Cen), telomeric (Tel), and split signal (On). **d** Quantification of chromosomal aberrations relative to the probe, as represented in (**c**). RP11-316D5 (*N* = 98); RP11-265L20 (*N* = 82); RP11-707K3 (*N* = 134). **e** RT profiles of Ctrl and APH-treated cell at the fragile site 22q12, as described in (**a**). Yellow box: fragility core within TTC28. **f** Bru-seq traces of Ctrl and APH-treated cells at 22q12, as described in (**b**), Annotated genes are shown at the bottom. **g** Representative images of chromosomal breaks on metaphase spreads stained with PI (red), treated with APH and labeled with the FISH probes (green), RP11-158G12 and RP11-61D21, and centromere 22 (cent., green) as described in (**c**). Breaks positioned relative to the probe: centromeric (Cen)—RP11-61D21; telomeric (Tel)—RP11-158G12. **h** Quantification of chromosomal aberrations relative to the probe, as represented in (**g**), RP11-158G12 (*N* = 44) to RP11-61D21 (*N* = 55). **d**, **h** *N* represents the number of analyzed chromosomes from two independent experiments. Scale bars: 2 μm. Source data are provided as a Source data file.

The 2q22 band spans >10 Mb and has three prominent late replicating regions in control cells (Fig. 2a), which are not affected by APH (Fig. 2a) and one mid-S replicating plateau region that is substantially delayed (Fig. 2a). Interestingly, this RT-delayed domain coincides with a large gene, *GTDC1* (386 kb, Fig. 2a). Moreover, flanking *GTDC1* are two large domains, which replicate earlier following APH treatment (Fig. 2a). Thus, APH

induced a V-shaped RT pattern at the *GTDC1* locus, which indicates that the region is replicated by single/few slow and long traveling forks with earlier activation of origins flanking the delayed region. This is in contrast to the RT plateau in control cells, implying that under normal conditions the replication fork speed is sufficient to properly replicate the gene. Similarly, the other fragile band, 22q12, also spans >10 Mb and has a prominent

late replicating domain, which is not affected by APH (Fig. 2e) and an early/mid-S domain which is delayed by APH (Fig. 2e). This delayed domain coincides with a large gene, *TTC28* (~700 kb), generating a V-shaped RT profile, that was not flanked by earlier replication in this case. Therefore, the APH-induced V-shaped RT profile may represent a high-resolution signature of fragility.

To test whether the APH-induced V-shaped RT profile is the CFR at 2q22 and 22q12 we mapped the fragility using FISH probes delimiting the V-shaped RT domains (Fig. 2a, e), and examined the position of breaks relative to the probes. The analysis showed that in 2q22 the vast majority of breaks occurred between probes RP11-316D5 and RP11-707K3, at the replication stress sensitive V-shaped RT domain marked by the RP11-265L20 probe (Fig. 2c, d), indicating that this locus is indeed the CFR. Moreover, the three prominent late replicating regions, which are not affected by APH, are excluded from the fragility core (Fig. 2a). Similarly, in 22q12, most breaks were found between the FISH probes (RP11-158G12 and RP11-61D21) delimiting the V-shaped RT domain, while the late replicating region, which is not affected by APH, is excluded from the fragility core (Fig. 2g, h). Altogether, the results demonstrate that the CFR is not located within the latest replicating regions at a CFS band, as previously suggested[20], but lies within a region replicating in early/mid-S phase that is delayed by APH. These results re-emphasize the importance of studying RT at CFSs under replication stress conditions.

**Delayed replication of large genes is insufficient to drive CFS expression.** CFSs are enriched with large genes (>300 kb)[42]. Therefore, to investigate the association between large genes, APH-induced RT delay and chromosomal fragility (Fig. 2), we analyzed the RT profiles of genes across the genome (data are binned in 5 kb nonoverlapping windows, thus excluding genes <5 kb). RT analysis showed that genes >250 kb tend to replicate later in S phase compared with the rest of the genes in control cells (Fig. 3a, Supplementary Fig. 5a, b, Supplementary Data 2), in agreement with previous reports[42]. Interestingly, this is also true for APH-treated cells (Fig. 3a, Supplementary Fig. 5a, b). Notably, the average RT of these genes was mid-S phase, as observed for the fragile genes in 2q22 and 22q12 (Fig. 2). Furthermore, the RT of only large genes was affected by APH (Fig. 3a, Supplementary Fig. 5a, b). Since CFSs are enriched with large genes (>300 kb)[42], we next investigated the association between gene size and APH-induced RT delay. We sub-grouped the large genes according to size into four groups. RT analysis showed that the larger the genes are the later they are replicated, both in control and APH-treated cells (Fig. 3b, Supplementary Fig. 5c, d). Moreover, this analysis confirmed that the RT of all large genes is delayed by APH (Fig. 3b, Supplementary Fig. 5c, d), implying that the delay is dependent on gene size but restricted to large genes.

Next, we investigated the relationship between delayed replication of large genes and chromosomal instability. We divided large genes into two groups, within or outside a cytogenetic band identified as a CFS in BJ-hTERT cells. The analysis showed that the average RT of large genes is mid-S phase for both groups of genes (Fig. 3c, Supplementary Fig. 6a, b). Moreover, the RT of these large genes is further delayed by APH (Fig. 3c, Supplementary Fig. 6a, b). RT delay was found for both groups, implying that delayed replication of large genes may be necessary but is insufficient to drive chromosomal instability under stress. Next, we analyzed the RT of all genes within or outside CFS bands to investigate whether the RT of smaller genes within CFSs may also be affected by APH. RT analysis showed that genes <250 kb are early replicating and are not affected by

APH (Supplementary Fig. 6c, d). Thus, RT delay is restricted to large genes whether or not they overlap a CFS band.

**APH-induced replication delay of large genes is associated with transcription.** Previously, expression of large genes was suggested to associate with chromosomal instability at several CFSs[25,43]. However, a recent study reported that the expression level of very large genes affects their stability under replication stress[44]. Therefore, we investigated the expression level of genes following APH treatment. For this we used Bru-seq, which maps nascent RNA transcripts, and allows measuring active transcription[45]. The analysis showed no APH effect on gene expression (Supplementary Fig. 7). Next, we analyzed the RT profile of large genes located within a CFS band, either expressed or silent. The analysis showed that the average RT of silent large genes is late S phase and is not affected by APH (Fig. 3d, Supplementary Fig. 8a). Interestingly, the average RT of expressed large genes in fragile bands is early S phase and is delayed by APH (Fig. 3d, Supplementary Fig. 8b). We then examined the RT profiles of large genes located outside of CFS bands. The analysis showed similar profiles to those of large genes within CFS bands (Supplementary Fig. 8c, d). These results further support the premise that delayed replication of large genes may be necessary but is insufficient to drive chromosomal instability under stress.

In light of the marked difference in the RT profiles of expressed and silent large genes, we examined the RT of all genes by their transcription profile. The analysis showed that expressed genes are earlier replicating compared with silent genes, as previously reported[1,46,47] (Supplementary Fig. 9a, b). Interestingly, APH-induced delayed RT was found only in expressed large genes (Supplementary Fig. 9b). We next explored the RT of large genes sub-grouped according to size and found that the larger the genes are the later they replicate both in control and APH-treated cells, whether expressed or silent (Supplementary Fig. 9c, d). However, expressed large genes are delayed whereas silent large genes are not affected. Overall, the RT of large genes seems to be more adjustable and correlated with transcription than the RT of smaller genes.

We then analyzed the RT of large genes divided by expression level quartiles. The results showed that large genes moderately (q3) or highly (q4) expressed were delayed by APH (Supplementary Fig. 10). In contrast, the RT profile of silent (q1) or weakly expressed (q2) genes was not affected (Supplementary Fig. 10). Altogether, these results suggest that for large genes transcribed above the median level, the RT and susceptibility to replication stress are transcription dependent.

**Delayed RT of expressed large genes is associated with chromosomal fragility.** In light of the transcription-associated RT delay in large genes (Fig. 3), we explored the expression profile of large genes within the CFS bands, 2q22 and 22q12 (Fig. 2). There are five large genes in 2q22: *ARHGAP15*, *GTDC1*, *LRP1B*, *TEX41*, and *THSD7B*, but only the *GTDC1* region was found to be fragile by FISH mapping (Fig. 2c, d). The *GTDC1* locus is delayed by APH (Fig. 2a) and moderately expressed (q3), in control and APH-treated cells (Fig. 2b). Interestingly, all other non-fragile large genes in the band are advanced by APH, whether silent (q1, *ARHGAP15*, *TEX41*, and *THSD7B*) or weakly expressed (q2, *LRP1B*) (Fig. 2c, d). Together these results suggest that large genes expressed above the median gene expression level are sensitive to replication stress which leads to fragility.

We further tested this conclusion in the fragile site at 22q12, in which the core fragility was mapped (Fig. 2e). There are three large genes at 22q12: *TTC28*, *SYN3*, and *LARGE1*. The fragile *TTC28* locus is delayed by APH (Fig. 2e) and highly expressed

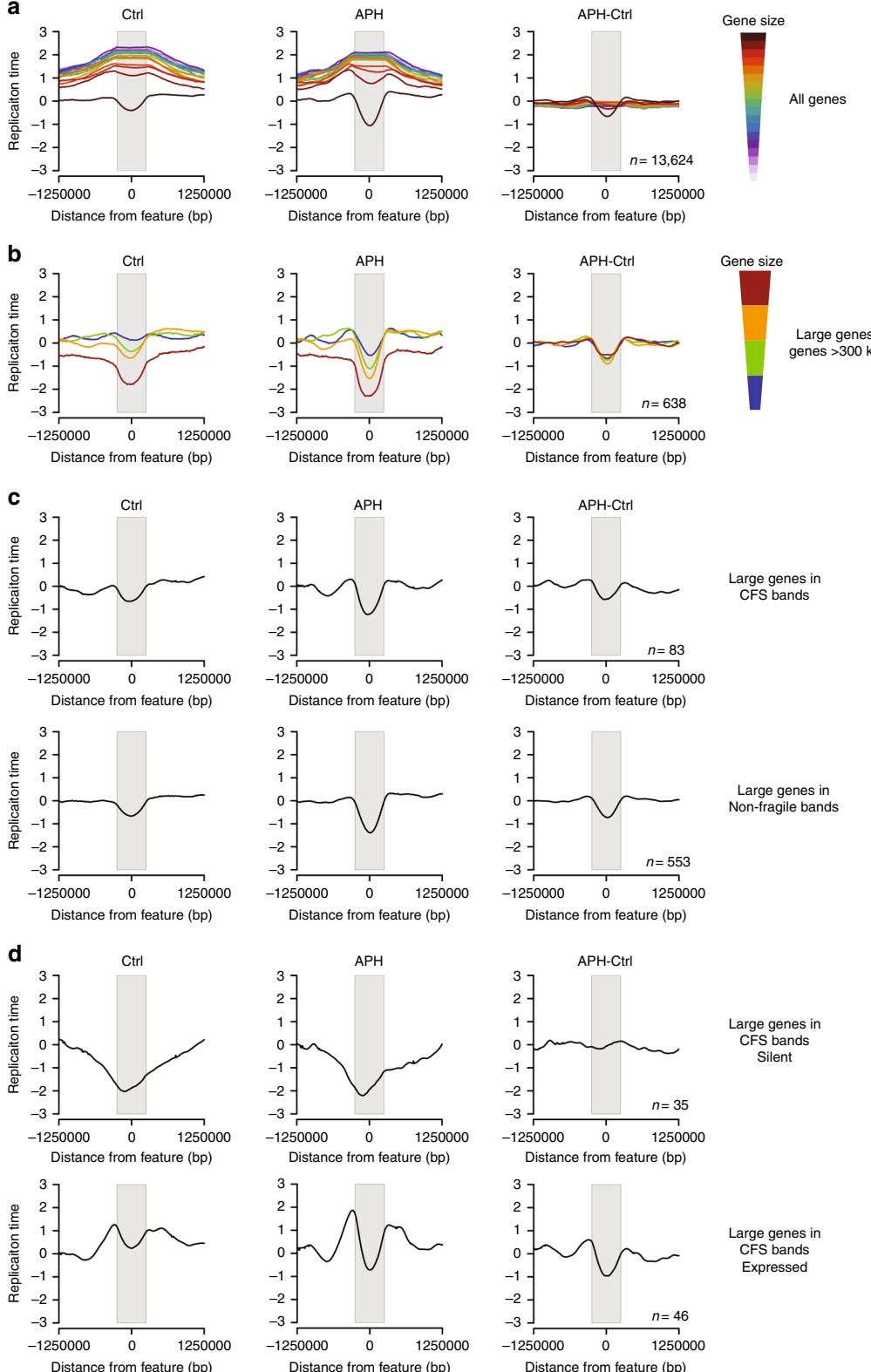

**Fig. 3 APH delays the RT of expressed large genes. a** Averaged RT of all genes clustered by size in Ctrl and APH-treated cells. The difference in RT following APH treatment is presented as the subtraction of the RT in Ctrl from the RT in APH-treated cells (APH-Ctrl). Genes are centered at 0 and stretched to fit the gray box, and 1 Mb upstream and downstream the genes are presented. Genes are sorted according to size from top to bottom (largest to smallest, respectively) and color coded by cluster as indicated. **b** Averaged RT of large genes (>300 kb) are represented as in (**a**). Genes are sorted according to size from top to bottom (largest to smallest, respectively) and color coded by cluster as indicated in Ctrl, APH, and APH-Ctrl. **c** Averaged RT of large genes within CFS bands (at the top) and in non-fragile cytogenetic bands (at the bottom) in Ctrl, APH, and APH-Ctrl. **d** Averaged RT of large genes within CFS bands. At the top are silent genes, at the bottom are expressed genes in Ctrl, APH, and APH-Ctrl.

(q4) in control and APH-cells (Fig. 2f). In contrast, both non-fragile large genes are not affected by APH, whether silent (*SYN3*) or weakly expressed (*LARGE1*) (Fig. 2e, f). These results show that not all expressed large genes are delayed by APH. Altogether, the results suggest that chromosomal fragility is induced at large genes, expressed above the median level, with delayed RT. However, whole genome transcription and RT analyses identified 119 regions with delayed RT within highly transcribed large genes but only 17% of them are within CFSs (Supplementary Data 3). Hence, these factors are necessary but insufficient to induce chromosomal instability.

**Chromosome stability is compromised at TAD boundaries.** The 3D architecture of the genome was found to associate with RT and transcriptional programs, and further suggested to regulate DNA-damage repair[29,33,36]. Thus, we investigated the effect of TAD organization on chromosomal fragility. We identified TADs based on the TAD-separation score for immortalized human fibroblasts (HFF-hTERT) from the 4D Nucleome project[48] and explored their association with APH-induced RT changes and transcription of large genes. First, we examined the CFRs of 2q22 and 22q12 (Fig. 2). We observed that the CFR of each of these FISH mapped CFSs spans a TAD boundary (Fig. 4a–f). Moreover, 20/24 (83%) cytogenetically mapped CFSs in BJ-hTERT cells[41], that harbor a RT-delayed transcribed large gene, span a TAD boundary. Importantly, only 2/24 (8%) cytogenetically mapped CFSs harbor a RT-delayed transcribed large gene located within a TAD domain, implying that the vast majority of CFSs span TAD boundaries harboring an APH-induced RT delay transcribed large gene.

Next we explored the RT profile of large genes either spanning a TAD boundary or located within a TAD domain (intra-TAD). The results showed a delay in RT by APH in both groups of large genes (Supplementary Fig. 11a, b). We analyzed the RT of TAD boundary overlapping and intra-TAD large genes according to their transcription state (silent or expressed) (Supplementary Fig. 12). The analysis showed that both intra-TAD and TAD boundary overlapping silent large genes are not delayed by APH (Supplementary Fig. 12a, c, e), whereas expressed large genes are delayed (Supplementary Fig. 12b, d). Interestingly, the delay in the averaged RT of TAD boundary spanning large and transcribed genes is significantly larger compared with the delay in intra-TAD transcribed large genes (Supplementary Fig. 12f). Altogether, the results indicate that TAD boundary spanning large expressed genes are more susceptible to replication delay under stress conditions than large expressed genes located within TADs.

Next, we explored whether TAD boundary architecture affects the RT program. For this we analyzed the RT profile around all TAD boundaries. The results showed no change in RT by APH, within or around TAD boundaries, implying that the TAD boundary per se does not affect the RT program (Supplementary Fig. 13a). We then explored the RT profile of TAD boundaries located within genes (Supplementary Fig. 13b). The results showed that only TAD boundaries located within large genes were delayed by APH (Supplementary Fig. 13b). Therefore, we further analyzed the RT of TAD boundaries in silent compared with expressed large genes (Supplementary Fig. 13c, d). The results showed that the RT of TAD boundaries within silent large genes was late replicating and was not affected by APH (Supplementary Fig. 13c). In contrast, the RT of TAD boundaries located within expressed large genes was early replicating and delayed (Supplementary Fig. 13d). Moreover, the TAD boundaries are positioned at the most delayed region within these genes. Altogether, the results raise the possibility that TAD boundaries within APH-delayed expressed large genes may perturb the completion of DNA replication in these regions, implying that the 3D architecture affects the RT of large expressed genes under replication stress.

We then analyzed the TAD boundary dispersion across all large genes. The analysis revealed that most large genes overlap a TAD boundary (442/638, 69%) (Supplementary Data 4). However, only 56/442 (13%) genes are located within cytogenetically mapped fragile sites in BJ-hTERT cells[41], supporting the conclusion that TAD boundary architecture by itself is insufficient to drive fragility. Interestingly, the majority (312/442, 71%) of the large genes spanning a TAD boundary is not delayed by APH and most (236/312, 75%) of these genes are silent or weakly expressed (q1 and q2) (Supplementary Data 4). Altogether, these results suggest that most large genes spanning TAD boundaries are not fragile, as they lack part of the fragility signature, either RT delay and/or active transcription.

TAD boundaries are enriched with CTCF binding[30,31]. However, analysis of CTCF binding across the genome, revealed that the vast majority of binding sites is not-at TAD boundaries[30,31]. Therefore, we tested whether CTCF plays a role in fragility. We analyzed publically available CTCF ChIP-seq data of HFF cells from the ENCODE project. The results showed indeed that TAD boundaries are enriched with CTCF binding (Supplementary Fig 14a, b). However, as previously reported, CTCF binds genome-wide, and thus the vast majority of binding sites are not associated with TAD boundaries (Supplementary Fig 14c). We then analyzed CTCF binding at large genes overlapping a TAD boundary and found that the majority of these genes (423/442, 96%) are bound by CTCF, as expected. However, most (182/196, 93%) of the large genes located within a single TAD domain were also bound by CTCF. Thus, CTCF probably does not play a significant role in fragility as most large genes are bound by CTCF, whether they are intra-TAD or spanning a TAD boundary.

Remarkably, the vast majority (95/119, 80%) of RT-delayed transcribed large genes is overlapping a TAD boundary (Supplementary Data 4). Out of them 20 were previously mapped cytogenetically as CFSs[41]. We, therefore, hypothesized that cytogenetic mapping, using G-banding, may have a limited resolution for detection of chromosomal lesions such as gaps at CFSs, leading to underestimation of chromosomal instability. Thus, we investigated the stability of loci with the fragility signature, that were not identified cytogenetically as CFSs. For this we performed FISH on six fragility signature candidate regions and four non-fragile intra-TAD regions with delayed RT of large transcribed genes, to test whether the chromosomal architecture underlies fragility (Fig. 4g–n, Supplementary Data 5). In addition, we FISH mapped two previously cytogenetically mapped CFSs[41] as a control (Fig. 4n, Supplementary Data 5). The analysis identified recurrent chromosomal fragility at all six fragility candidate regions, spanning a TAD boundary (Fig. 4m, n, Supplementary Fig. 15, and Supplementary Data 5). In contrast, no recurrent breaks were found at the four intra-TAD regions although they harbor RT-delayed transcribed large genes (Fig. 4j–n, Supplementary Data 5). It is worth noting that only 2/24 RT-delayed intra-TAD transcribed large genes are located within cytogenetically mapped CFS bands, suggesting that fragility is not associated with intra-TAD architecture. Analysis of the known CFSs identified recurrent breaks at these loci, as expected (Fig. 4n, Supplementary Fig. 15, and Supplementary Data 5). Overall, the results highlight the effect of TAD organization on fragile site stability, implying that intra-TAD regions are safeguarded from deleterious replication stress, whereas inter-TAD regions in combination with high transcription and RT delay are susceptible to breakage.

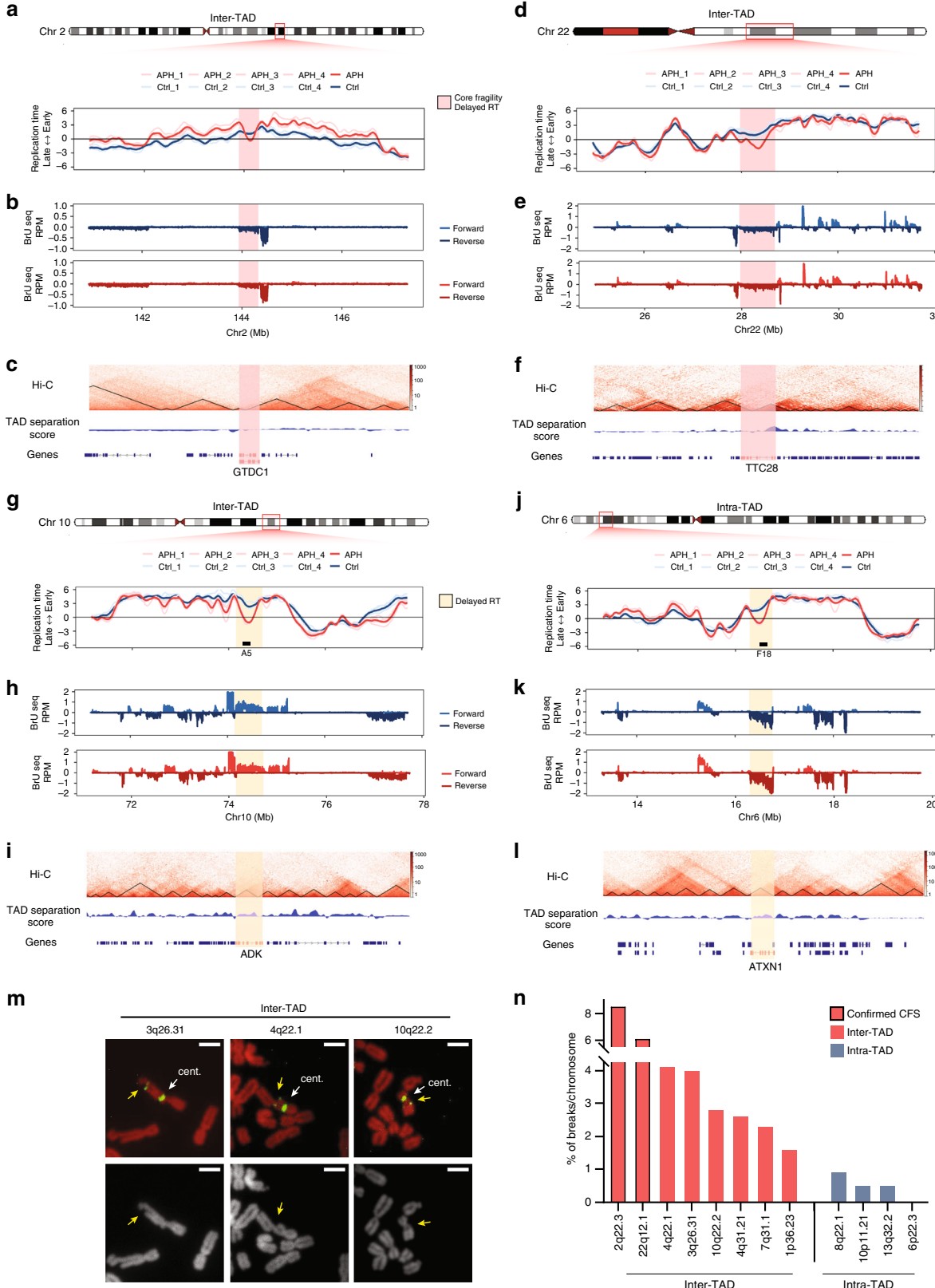

## Discussion

Here we have revealed a signature for chromosomal fragility upon replication stress, comprised of TAD boundary spanning actively transcribed large genes with APH-induced delay in RT. This combined multilayer signature reveals the underlying mechanism of CFS expression linking the 3D organization of regions harboring transcribed large genes with altered RT.

One necessary but insufficient component of the revealed fragility signature is delayed RT. DNA replication in eukaryotes is a temporally and spatially orchestrated process[1]. The RT program is tissue specific and is associated with chromatin architecture and

**Fig. 4 Inter-TAD compromises RT delayed highly expressed large genes' stability under replication stress.** RT profiles and nascent RNA transcription of Ctrl and APH-treated cells at the 3 Mb flanking the CFR of mapped CFSs: 2q22 (**a**, **b**) and 22q12 (**d**, **e**), as described in Fig. 2. The red boxes indicate the CFR within each CFS, mapped in Fig. 2, overlapping a large, highly transcribed and RT-delayed gene. **c**, **f** Hi-C heat maps and TAD-separation scores of control HFF-hTERT cells at the CFSs: 2q22 (**c**) and 22q12 (**f**). **g–l** RT profiles, nRNA transcription and Hi-C heat maps (as described in **a–c**) of the 3 Mb flanking a candidate overlapping a TAD boundary (inter-TAD) fragile site on chromosome 10q22 (**g–i**) and an intra-TAD control region on chromosome 6p22 (**j–l**). FISH probes are presented as black boxes. Annotated genes are shown at the bottom. Yellow boxes mark large, transcribed genes with RT delay, *ADK* (**i**) and *ATXN1* (**l**). **m** Representative images of breaks on metaphase spreads from BJ-hTERT cells treated with APH, quantified in (**n**). Top panel: propidium iodide (PI, red) staining of chromosomes, yellow arrows mark breaks and gaps at FISH probes (left to right: RP11-354I1 (3q26.31), RP11-706L3 (4q22.1), and RP11-98A5 (10q22.2)) and white arrows mark the centromeres (cent.) of the matching chromosome. Bottom panel: contrast of PI staining, yellow arrows mark breaks and gaps. **n** Quantification of the percent of breaks and gaps out of total number of relevant chromosomes analyzed from two independent experiments (for detailed quantification please see Supplementary Data 5). In red are inter-TAD loci, out lined in black are confirmed CFSs, and in blue are intra-TAD loci. Scale bars: 2 μm. Source data are provided as a Source data file.

transcriptional activity[1]. Previous replication dynamics analyses of single DNA fibers showed that APH-induced replication stress leads to fork slowing/stalling, which subsequently activates dormant origins in an attempt to compensate for the perturbed fork progression[6]. Here, we show that mild replication stress induced by APH alters the temporal order of replication of only a small part of the genome (~4% of the genome), which is highly enriched for CFSs (Fig. 1). This suggests that dormant origin activation by APH is sufficient to preserve the normal and scheduled RT program in the bulk of the genome, as also implied by the pulse-chase-pulse assay (Supplementary Fig. 1). It is interesting to note that replication stress induced by APH advanced the RT of certain genomic regions (Fig. 1d), presumably by earlier origin activation, suggesting that in addition to a local activation of dormant origins in response to replication stress, in some genomic regions there is a regional response leading to earlier origin activation. It is tempting to speculate that such advanced RT may play a protective role in maintaining chromosomal stability under replication stress as most advanced RT regions are not within the cytogenetically mapped CFSs, and advanced regions within the mapped CFSs are outside the CFR (Fig. 2).

Previous studies investigating chromosomal instability at several CFSs reported that the fragility core within these CFSs corresponded to the latest replicating regions which are origin poor, as shown by analyses of replication dynamics, RT and ORC2 ChIP-seq under normal conditions[20,21,49]. These results suggest that paucity of origins in the latest RT regions determines the replication profile at CFSs and thus underlies the failure of these regions to complete replication upon replication stress. It is worth noting that these RT analyses were performed on cells grown under normal conditions whereas our analyses of the RT profiles were performed under both normal and replication stress conditions. Our results, however, revealed that most of the fragile site regions replicate in early/mid-S phase in control conditions and are delayed under replication stress, in agreement with a recent study of CFS replication timing[50]. This suggests that the normal RT is not the main factor underlying chromosomal fragility. Indeed, most late replicating regions under normal conditions are not fragile. The delayed RT identified in cells treated with APH revealed a replication stress-induced V-shaped RT domain most likely resulting from slowed fork progression and lack of origin firing. These findings are in agreement with a previous model in which the fragility core at CFSs is origin deficient[20,21,49–51]. However, not all RT-delayed loci are fragile, indicating that sensitivity to replication stress may be necessary but insufficient to drive fragility. It should be noted that the RT data was binned into 5 kb windows, thus, we cannot identify smaller regions which may remain under-replicated until mitosis due to perturbed replication.

CFSs were found to be enriched with large expressed genes[24,25,42,43]. Our analyses revealed that CFSs harbor delayed RT regions which coincided with large genes. Moreover, the RT analysis showed that only the RT of large genes is delayed by APH, as compared with the rest of all annotated genes. Further, separating large genes into transcriptionally silent or active showed that highly transcribed large genes are early replicating under normal conditions and only their RT is delayed by APH (Fig. 3, Supplementary Fig. 9). These results are in agreement with a recent study showing preferential replication initiation at transcription start sites of highly expressed genes[52]. Under replication stress, the replication initiated downstream of genes[52]. Thus, transcription of large genes may promote the RT delay under replication stress, by deploying licensed origins at the flanking regions of genes and/or by promoting transcription–replication collisions within the gene bodies, in a non-mutually exclusive manner, as previously suggested[25,53]. Our analyses revealed, however, that large highly expressed genes with RT delay are also found in non-fragile bands, emphasizing that transcription and RT delay are necessary but insufficient factors leading to fragility.

Eukaryotic nuclei are organized into TADs that unite loci sharing a common function. TADs are essential for coordinated replication, transcription, and DNA-damage repair[29,33,36]. Damage repair might be preferentially confined to intra-TAD regions, as shown by a tendency of γH2AX spreading to stop at TAD boundaries[36,39]. Recently, the DNA structure-specific nuclease MUS81-EME1 was shown to localize to CFSs in early mitosis[54,55], and cleave the under-replicated DNA, triggering DNA repair synthesis in mitosis[11]. Our Hi-C analysis revealed that CFSs are enriched with large highly expressed genes with RT delay spanning TAD boundaries. This suggests that CFS stability may be compromised by lack of DNA repair in the TAD boundaries at the CFRs leading to the transmission of damage to daughter cells and propagation of genomic instability[56]. Interestingly, the averaged RT of both TAD boundary spanning and intra-TAD domain actively transcribed large genes is delayed by APH. However, the RT delay of TAD boundary spanning genes is significantly larger, implying perturbed DNA replication at TAD boundaries overlapping large expressed genes (Supplementary Fig. 12). This is supported by the finding that TAD boundaries are at the core delayed region within expressed large genes (Supplementary Fig. 13d). It is worth noting that many large genes spanning a TAD boundary are not fragile as they lack either RT delay and/or active transcription, implying that TAD boundary architecture by itself, although required, is insufficient to underlie fragility.

Genome-wide screen for this fragility signature under replication stress identified 95 regions, of which 20 were previously identified cytogenetically as CFSs in the same cells[41]. Fragile sites appear as gaps and breaks in metaphase chromosome from cells grown under replication stress. Cytogenetic mapping using G-banding allows the identification of chromosomal breaks,

however, gaps are difficult to observe. We therefore, speculated that the limited fragility resolution of G-banded chromosomes leads to underestimation of the fragility landscape. Indeed, our FISH analysis of TAD boundary spanning regions showing delayed RT in highly transcribed genes found recurrent gaps/breaks, under APH treatment (Fig. 4g–n and Supplementary Fig. 15), as predicted by the fragility signature. Importantly, intra-TAD regions, with RT delayed in highly transcribed large genes did not show recurrent gaps/breaks. These results suggest an important role of chromatin conformation in underlying chromosomal instability at fragile sites.

Altogether, the results presented here suggest that the molecular basis underlying genomic instability at CFSs is a multilayer combination of replication, transcription and genome 3D organization (Supplementary Fig. 16). This fragility signature enabled precise mapping of the CFR and identification of novel fragile sites, not detected cytogenetically, highlighting the improved sensitivity of our approach for identifying fragile sites. Moreover, as fragile sites are cell-type specific[42,57,58] revised mapping of fragile sites using RT profiling in cells grown under normal and replication stress conditions, should be performed for identification of the entire repertoire of fragility in different cell types. Our results are relevant for understanding genomic instability in cancer as cancer development is promoted by replication-induced genomic instability[4,7,59], changes in gene expression[60] and 3D disorganization of the genome[61].

## Methods

**Cell culture and treatments**. Telomerase-immortalized human foreskin fibroblasts BJ-hTERT cells (Clontech) were grown in DMEM supplemented with 10% fetal bovine serum, 1,000,000 U l$^{-1}$ penicillin, and 100 mg l$^{-1}$ streptomycin. The cells were tested and found negative for mycoplasma. Cells were treated with 0.2 μM aphidicolin and 0.73–1.46 mM caffeine in growth media for 24 h prior to fixation.

**FISH mapping on metaphase chromosomes**. Cells were treated with 100 ng ml$^{-1}$ colcemid (Invitrogen, Carlsbad, CA, USA) for 40 min in incubator at 5% CO$_2$. Then, cells were collected by trypsinization, treated with hypotonic solution at 37 °C for 30 min and fixed with multiple changes of methanol:acetic acid 3:1. Fixed cells were kept at −20 °C until analysis. For analysis of total gaps, constriction or breaks chromosomes were stained with propidium iodide and blindly analyzed. For fluorescent in situ hybridization (FISH) analysis, BAC clones (RP11-316D5, RP11-265L20, RP11-158G12, RP11-61D21, RP11-707K3, RP11-354I1, RP11-706F7, RP11-98A5, RP11-622F6, RP11-830A20, RP11-346F18, RP11-16G14, RP11-401D21, RP11-192I20, RP11-706L3) were labeled with by nick translation. In order to evaluate metaphases which experienced similar level of replication stress, metaphases that exceeded a threshold of 5 breaks were not included in the FISH evaluation.

**Genome-wide RT profiling**. Genome-wide RT profiles were constructed as previously described[62]. Briefly, cells were pulse labeled with BrdU, separated into early and late S-phase fractions by flow cytometry, and processed by Repli-seq. Sequencing libraries of BrdU-substituted DNA from early and late fractions were prepared by NEBNext Ultra DNA Library Prep Kit for Illumina (E7370; New England Biolabs). Sequencing was performed on an Illumina-HiSeq 2500 sequencing system by 50-bp single-end reads. Reads with quality scores above 30 were mapped to the Hg38 reference genome using bowtie2. Approximately 8 million uniquely mapped reads were obtained from each library. Read counts were binned into 5-kb nonoverlapping windows, and log2 ratios of read counts between early and late fractions were calculated. Plots of RT profiles were generated in R project for Statistical Computing (htpp://www.r-project.com). Heat maps and averaged RT profiles of genes clustered by size were generated using genmat package for R (https://rdrr.io/github/dvera/genmat/). Briefly, genes were centered at the mid gene body indicated by 0 and stretched to the same length indicated by gray boxes. For non-stretched figures, genes were only centered at the mid gene body.

**RT signatures identification**. RT profiles were expressed as numeric vectors. A threshold of two standard deviations in pairwise comparisons of technical repeats (≥1 RT values) was used to identify nonoverlapping and variable regions (100-kb windows) in pairwise comparisons between all samples. Unsupervised hierarchical and k-means clustering analysis were performed using Cluster 3.0[63]. Heat maps and dendrograms were generated in R project for Statistical Computing (htpp://www.r-project.com). To demonstrate the statistical significance of the RT changes identified we performed a pairwise t-test per RT signature.

**Bru-seq**. Bru-seq was performed as previously described[45]. Briefly, bromouridine (Bru) (Aldrich) was added to the media to a final concentration of 2 mM and incubated at 37 °C for 30 min. Total RNA was isolated using TRIzol reagent (Invitrogen), and Bru-labeled RNA was isolated by incubation of the isolated total RNA with anti-BrdU antibodies (BD Biosciences) conjugated to magnetic Dynabeads (Invitrogen) under gentle agitation at room temperature for 1 h. cDNA libraries were prepared from the isolated Bru-labeled RNA using the Illumina TruSeq library kit and sequenced using Illumina-HiSeq sequencers at the University of Michigan DNA Sequencing Core. The sequencing and read mapping were carried out as previously described[45].

Reads were first aligned to the ribosomal RNA (rRNA) repeating unit (GenBank U13369.1) and the human mitochondrial and EBV genomes using Bowtie2 (2.3.3). Reads that did not align to rRNA/chrM/EBV were mapped to hg38 using STAR (v 2.5.3a)[64] and a STAR index created from GENCODE annotation version 27[65]. RPM-normalized read densities were calculated with bedtools2[66] genomecov, using a scaling factor of 1,000,000/(number of parsed reads in library). The profiles are shown at 1 kb window bins.

Differential expression analysis was performed using DESeq2 (1.6.3)[67] with betaPrior set to False.

**S-phase kinetics assay**. Cells were treated with APH at described above for 24 h prior to fixation. Four hours prior to fixation cells were treated with 10 mM IdU for 30 min, washed twice with PBS and chased for 3 h. After the 3 h chase interval, cells were labeled with 10 mM EdU for 30 min. Cells were fixed with 4% formaldehyde for 15 min, permeabilized with 0.5% Triton X-100 for 15 min and Click-iT reaction was performed (Invitrogen). Afterward, cells were incubated with a primary antibody for fluorescence detection of IdU, mouse anti-BrdU (Becton Dickinson) overnight, and the secondary antibody was goat anti-mouse Alexa Fluor 488 (Invitrogen). Images were analyzed double blindly using Fiji[68].

**Hi-C**. Raw Hi-C data of immortalized human foreskin fibroblasts (HFF-hTERT) were downloaded from the 4DN project (4DNESB6MNCFE)[48]. Read mapping, contact matrices generation, TAD calling and the identification of TAD boundaries were carried out as previously described by HiCExplorer[69].

**CTCF ChIP-seq analysis**. The CTCF ChIP-seq data of HFF cells from the ENCODE project (GSM1022644) were used. For expected CTCF binding at TAD boundaries, 1000 iterations of random size matching "TAD boundaries" were performed using bedtools shuffle –noOverlapping, excluding observed TAD boundaries from hg38.

**Reporting summary**. Further information on research design is available in the Nature Research Reporting Summary linked to this article.

## Data availability

All sequencing files and processed count matrices were deposited in Gene Expression Omnibus (GEO) under accession numbers (GSE150354 and GSE150543). Previously published Hi-C data of immortalized human foreskin fibroblasts (HFF-hTERT) were downloaded from the 4DN project (4DNESB6MNCFE). Previously published CTCF ChIP-seq data of HFF cells were downloaded from the ENCODE project (GSM1022644). All data are available from the authors upon reasonable request. Source data are provided with this paper.

## Code availability

The computer codes are available on GitHub (https://github.com/dvera/genmat/). Any further data are available from the authors upon reasonable request.

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

## Acknowledgements

This research was supported by grants to B.K. from the Israel Science Foundation (grant No. 176/11), the Israel Science Foundation (grant No. 1284/18), the Israeli Centers of Research Excellence (I-CORE), Gene Regulation in Complex Human Disease, Center No. 41/11, the ISF-NSFC joint program (grant No. 2535/16), and to D.M.G. from the NIH grant GM083337. The authors thank the members of the Kerem lab for thoughtful

discussions and advice, Dr. Daniel Vera for bioinformatics analyses, and Dr. Noemie Stanleigh for review and comments on the paper.

## Author contributions

D.S. contributed to conception and design, performed experiments, collection and assembly of data, data analysis and interpretation, and manuscript writing; T.S. performed the RT experiments and analyzed the data; M.I.T.S. performed the FISH experiments and analyses; K.M. contributed to the conception and design, participated in performing the RT, nascent RNA sequencing, and FISH analyses; J.C.R.M. performed the RT signature analyses; B.M. performed nascent RNA-sequencing analyses; M.L. contributed to the nascent RNA-sequencing design and analyses; D.M.G. contributed to the design of the RT experiments, design and interpretation, manuscript writing; B.K. contributed to conception and design, financial support, data analysis and interpretation, manuscript writing and final approval of the paper.

## Competing interests

The authors declare no competing interests.
