## [Peer Review File · Nature Communications]

Reviewers' comments:

Reviewer #1 (Remarks to the Author):

This article explores a number of factors that contribute to genome instability at common fragile sites (CFS). Despite much research, the factors that contribute to genomic instability at CFS remain somewhat elusive. The authors employ a methodical approach to assess the contribution of gene length, transcriptional level, alterations in replication timing and finally 3D chromosome structure to fragility of these sites in immortalized fibroblast cells. Their results reveal that one of the contributing factors for fragility is the shift in replication timing after mild replication stress, not late replication per se. In accordance with the literature, the authors also find that the fragile sites are associated with highly transcribed, long genes. Furthermore, the authors make a compelling case that the location of the long, highly transcribed genes within TAD domains contributes to their fragility. This is a very novel finding that merits publication.

I find the study to be well-designed, with solid conclusions that contribute to our understanding of fragile sites and to the field of cancer research in general. For this reason, I highly recommend the publication of this article in your journal.

I do however, have a few reservations that need to be addressed before the acceptance of the paper, which are listed below.

Major points

1. Could the authors please provide a more detailed explanation of how the RT-variable regions were obtained? Given that repli-seq is a complex analysis, I am tempted to accept having a fixed threshold to call RT-variable regions. However, I wonder if this analysis will stand multiple testing or other statistical significance tools usually employed for differential gene expression or differential binding analysis for ChIP-seq. These analyses consider that there are variations in the data from sample to sample and that not all variation is significant.

Could the "advance" RT profile be associated with this kind of random variation between samples? Could "earlier replication upon replication stress" be insignificant/artefact?

Our current understanding of ATR/ATM signalling upon replication fork stalling only points to delaying of origin firing and not advancing of origin firing. Thus, I believe the concept of "earlier replication upon replication stress" needs to be supported with clear, statistically significant data.

2. Figure 3. "implying that the APH-induced RT delay is dependent on gene size but restricted to large genes". While I agree with the authors that this is likely to be the case, it is hard to know how the authors normalised this. Obviously, larger genes (i.e. 300 kb) are more likely to contain RT-delay regions, simply because an 8 kb region (average gene size) is much less likely to contain an RT-delay region that is calculated in bins of 100 kb.

Can the authors provide equal size windows of regions containing (i) small or (ii) large genes and directly compare the RT profiles on these equal size windows?

3. related to the point above; Figure 3a: Can the authors please provide how exactly the profiles were drawn? From my understanding, all the genes are "stretched" to be the same size and fit the grey box (deeptools computeMatrix scale-regions?). If so, how would this figure look like if the authors were to use a reference point (i.e. gene center) without stretching of the genes?

4. Figure 3d: What does this exact figure look like for long genes that are not fragile sites? (silent vs transcribed). In other words, is the impact of transcription specific for long transcribed genes that are fragile? or do silent vs transcribed long genes that are not fragile also display the same profile?

5. Figure 4n: One of the most novel findings of this article hinges on this figure. Thus, I believe

this itneeds more samples to reach solid conclusions. Given the % of breaks observed (1-4%) Perhaps n=6 for fragile and at least n=4 for control regions, with at least 2 known fragile sites as positive controls are needed.

6. Repli-seq suggests that many common fragile sites are NOT late-replicating. However, due to its resolution (100 kb windows), Repli-seq cannot identify shorter regions (i.e. 1-2 kb) that may remain unreplicated until G2/M. This is a limitation of the assay and needs to be commented on in the Discussion.

Minor points

1. Figure 1b: As the authors state, the differences between the profiles are very small. In order to understand the sample distances a bit more clearly, is it possible that the authors provide a principal component (PCA) analysis for the 8 samples?

2. The FISH assays are performed on BAC clones. Is there literature on whether BAC clones form TADs and whether the TADs they form resemble their endogenous TADs?

3. Please also discuss factors other than what was studied in this article that might contribute to fragility. CFS have also been linked to "origin seeding" (PMID: 21258320). This was also suggested by both Miotti et al (ORC2 density) PMID: 27436900.

4. As a final figure and to understand the contribution of the identified factors to fragility, I would suggest to include what percent of intra-TAD regions harbouring highly transcribed long genes with RT-delay are fragile sites?

5. Please order the figures- supplemental 8 is mentioned right after supplemental 4?

6. Could it be a mistake that the authors wrote they have used Bowtie to map BRU-seq reads? While I understand that BRU-seq maps nascent transcription, in a window of 30 mins (labelling), intron splicing would clearly take place. Bowtie does not map exon-exon junctions; for genes with small exons this means many reads would not be mapped. STAR (or any aligner that maps junctions) would have been appropriate choices for RNA-sequencing data. If this was not a mistake, can the authors show that quantification of gene expression is comparable when using STAR/tophat or bowtie?

Reviewer #2 (Remarks to the Author):

In this manuscript Sarni et al, set to investigate the molecular mechanisms that drive common fragile sites (CFS) instability. They monitored gene expression and replication timing (RT) in the presence of aphidicolin, known to induce CFS expression, in human fibroblasts and further crossed their results with previously published Hi-C. By doing so they uncovered that CFS are very well predicted when considering the size of the gene, its expression level, its delayed RT after replication stress and its position regarding TAD boundaries: in other words they propose that large expressed genes, that undergo delayed RT after aphidicolin treatment, and that overlap with a TAD boundary define the molecular signature of CFS.

This is an exciting study, very clearly written, presenting robust data and that is of major interest for the field. I have few suggestions that in my opinion would improve the manuscript, and a couple of recommendations for a better presentation of the results.

While the relationship between gene size, gene expression, RT delay and fragility is very well developed, I found the last part on the relationship with TAD boundaries, which is also the most novel, a little bit underdeveloped. Could the authors provide additional analyses regarding this

point? (below some suggestions)

- For instance, could they show as performed in Fig. 3 and Sup Figures 4-6, the heatmap and average RT profiles, for the long genes overlapping a boundary or not, for the long expressed genes overlapping a boundary or not etc...
- Could they draw replication timing around all TADs boundaries? Around TAD boundaries located in large active genes ? etc...
- One would expect more CTCF binding at these CFS. Using already published dataset for CTCF distribution, could the authors show the CTCF enrichment on expressed large genes that are fragile, compared to expressed large gene that are not fragile? And on large/ expressed/RT-delayed/ fragile versus large/ expressed/RT-delayed/ non-fragile?. Another quantitative value that might be useful is the insulation score, which helps to define TAD boundaries.
- Fig3d, shows that CFS can also take place on large, silent, genes, that do not display changes in RT: does this specific category overlap TAD boundaries?, can the authors show the average CTCF enrichment compared to other large, silent, not fragile genes?

Minor points

- In Fig. 3 , Fig S4, Fig. S6 the legend for gene size goes to red (dark pink?) for very large gene to red for very small genes: I would recommend changing the color code to really see differences between large and small
- For all the heatmaps please make sure that the scale for RT timing data between Aphidicolin treated and control cells is the same.
- I would recommend changing the term "inter-TAD" as this is very confusing. Maybe something like "overlapping TAD boundary", or "TAD boundary containing genes"
- Typo in the abstract first line: "Common fragile sties", please change
- For the FISH experiment, can you indicate how many events were scored? I would also recommend explaining a little bit more Fig2c-d (on p8)
- Typo on Fig. S7 in the legend "No chnage"

Below please find a point by point response to all comments raised by the reviewers. Our replies are marked in blue:

Reviewers' comments:

Reviewer #1 (Remarks to the Author):

This article explores a number of factors that contribute to genome instability at common fragile sites (CFS). Despite much research, the factors that contribute to genomic instability at CFS remain somewhat elusive. The authors employ a methodical approach to assess the contribution of gene length, transcriptional level, alterations in replication timing and finally 3D chromosome

structure to fragility of these sites in immortalized fibroblast cells. Their results reveal that one of the contributing factors for fragility is the shift in replication timing after mild replication stress, not late replication per se. In accordance with the literature, the authors also find that the fragile sites are associated with highly transcribed, long genes. Furthermore, the authors make a compelling case that the location of the long, highly transcribed genes within TAD domains contributes to their fragility. This is a very novel finding that merits publication.

I find the study to be well-designed, with solid conclusions that contribute to our understanding of fragile sites and to the field of cancer research in general. For this reason, I highly recommend the publication of this article in your journal.

I do however, have a few reservations that need to be addressed before the acceptance of the paper, which are listed below.

Major points

1. Could the authors please provide a more detailed explanation of how the RT-variable regions were obtained?

Given that repli-seq is a complex analysis, I am tempted to accept having a fixed threshold to call RT-variable regions. However, I wonder if this analysis will stand multiple testing or other statistical significance tools usually employed for differential gene expression or differential binding analysis for ChIP-seq. These analyses consider that there are variations in the data from sample to sample and that not all variation is significant.

Could the “advance “ RT profile be associated with this kind of random variation between samples? Could “earlier replication upon replication stress” be insignificant/artefact?

Our current understanding of ATR/ATM signalling upon replication fork stalling only points to delaying of origin firing and not advancing of origin firing. Thus, i believe the concept of “earlier replication upon replication stress” needs to be supported with clear, statistically significant data.

We thank the reviewer for the comment. Replication timing (RT) data is distinct from differential gene expression or ChIP-seq analysis in terms of its dynamic range and degree of variation. Therefore, identification of RT variation across samples requires a different approach.

We have developed distinct methods to identify and validate the significance of RT changes (Ryba et al., 2011, PLoS Comp Biol, doi: 10.1371/journal.pcbi.1002225; Rivera-Mulia et al., 2015, Genome Res, doi: 10.1101/gr.187989.114.; Rivera-Mulia et al., 2017, PNAS, doi: 10.1073/pnas.1711613114; Rivera-Mulia et al., 2019, Blood Adv, doi:10.1182/bloodadvances.2019000641). We have demonstrated that RT is a highly robust chromatin property with very little variation among technical replicates (which is consistent with the strong correlation values showed in Fig. 1b). Thus, a threshold above the variation between technical replicates followed by an unsupervised clustering analysis is the best approach to identify and classify all variation across distinct samples. Nevertheless, we agree with the reviewer that a more detailed explanation of the method and how we define statistical significance was missing. Thus, we revised the Methods section in our manuscript (please see page 25, lines 11-18). Briefly, we established a threshold of two standard deviations in pairwise comparisons of technical repeats (≥ 1 RT values). We then calculated the variation between control and APH, this analysis validated our threshold, and confirmed that differences ≥ 1 RT unit in pairwise comparisons between all samples capture all variation between samples while removing variation between technical replicates. Results from this analysis have been included in a new figure, Supplementary Fig. 2a, of the revised manuscript. Additionally, this analysis also confirms the small degree of “random variation” between samples.

As requested by the reviewer in order to demonstrate the statistical significance of the identified RT changes we performed a pairwise *t*-test for each RT signature. Results from this analysis have been added to the new Supplementary Fig. 2b. The results show that the RT profiles of control and APH treated cells significantly differ in the identified RT signatures. This also indicates that following APH treatment advanced RT is not an artifact but rather a cellular response to replication stress. As the reviewer pointed out this earlier replication timing under replication stress is indeed an unexpected and interesting result. Further studies, which are beyond the scope of this manuscript are required to reveal the mechanism leading to this change in the RT regulation.

2. Figure 3. “implying that the APH-induced RT delay is dependent on gene size but restricted to large genes”. While I agree with the authors that this is likely to be the case, it is hard to know how the authors normalised this. Obviously, larger genes (i.e. 300 kb) are more likely to contain RT-delay regions, simply because an 8 kb region (average gene size) is much less likely to contain an RT-delay region that is calculated in bins of 100 kb.

Can the authors provide equal size windows of regions containing (i) small or (ii) large genes and directly compare the RT profiles on these equal size windows?

We thank the reviewer for the comment, which highlighted to us the need for a clearer description of the data analysis. We only binned the genome into 100 kb windows in the analysis presented in Fig 1d, in which we identified the RT variable regions. Importantly, all our other analyses were performed using RT values at intervals of 5 kb non-overlapping windows, including the analysis presented in Figure 3. We chose to analyze the RT of genes >5 kb in length, in order to avoid the potential obstacle mentioned by the reviewer.

In order to emphasize that the analyses were performed using 5 kb non-overlapping windows (as described in the Methods section) we revised the text in the Result section addressing the 5 kb resolution (page 7 lines 6-7 and page 10 line 20).

3. related to the point above; Figure 3a: Can the authors please provide how exactly the profiles were drawn? From my understanding, all the genes are “stretched” to be the same size and fit the grey box (deeptools computeMatrix scale-regions?). If so, how would this figure look like if the authors were to use a reference point (i.e. gene center) without stretching of the genes?

As requested by the reviewer we now extended the description of how the plots were generated in the Methods section and in the relevant figure legends (please see the Method section, page 25 lines 6-10). Briefly, heat maps and averaged RT profiles of genes clustered by size were generated using the `matHeatmap` function of the `genmat` package for R (<https://rdrr.io/github/dvera/genmat/>). Genes were centered at the mid gene body indicated by 0 and stretched to the same length indicated by grey boxes as shown in Figure 3a.

As further requested by the reviewer we generated additional figures in which genes were centered at the mid gene body but not stretched. We provide in the revised manuscript the RT profiles of non-stretched all genes and for the non-stretched large genes (please see new Supplementary Figures 5b and 5d and their legends).

4. Figure 3d: What does this exact figure look like for long genes that are not fragile sites? (silent vs transcribed). In other words, is the impact of transcription specific for long transcribed genes that are fragile? or do silent vs transcribed long genes that are not fragile also display the same profile?

As requested by the reviewer we added heat maps of RT profiles of silent and expressed large genes located in non-fragile cytogenetic bands (non-fragile), please see Supplementary Figure 8 and Results section, page 12 lines: 18-23. As can be seen in the figure, only expressed large genes (but not silent large genes) are delayed by APH, whether they are located in fragile or non-fragile cytogenetic bands. Thus, the impact of transcription on RT of long genes is independent of their cytogenetic location in fragile or non-fragile cytogenetic bands.

5. Figure 4n: One of the most novel findings of this article hinges on this figure. Thus, I believe this it needs more samples to reach solid conclusions. Given the % of breaks observed (1-4%) Perhaps n=6 for fragile and at least n=4 for control regions, with at least 2 known fragile sites as positive controls are needed.

As requested by the reviewer we extended the analysis shown in Fig 4n by testing additional regions. This includes: (A) Two additional regions with the fragility signature, one on chr4q22 and the other on chr1p36 (Fig. 4n, Supplementary Fig. 15 and Supplementary Table 5). Thus, the entire analysis includes now 6 regions with the entire fragility signature. All these regions showed recurrent instability under replication stress. (B) Two additional controls, that harbor large expressed genes with RT delay but are located within a single TAD. Thus, as suggested by the reviewer, the entire analysis includes now 4 control regions, all of which showed no-

recurrent fragility (Fig. 4n and Supplementary Table 5). (C) Two known CFSs as positive controls, located at chr2q22 and chr22q12 that showed recurrent fragility as expected (Fig. 4n, Supplementary Fig. 15 and Supplementary Table 5). Altogether, the entire analysis of the 12 regions confirm our initial results and solidify our conclusion that inter-TAD expressed and delayed large genes are susceptible to replication stress and are manifested as fragile sites. Figure 4n and the text (Results section, page 18) were modified accordingly.

6. Repli-seq suggests that many common fragile sites are NOT late-replicating. However, due to its resolution (100 kb windows), Repli-seq cannot identify shorter regions (i.e. 1-2 kb) that may remain unreplicated until G2/M. This is a limitation of the assay and needs to be commented on in the Discussion.

We thank the reviewer for the comment. We would like to emphasize as explained above that the RT data presented throughout the manuscript is binned into 5 kb non-overlapping windows and not 100 kb windows (except for RT signatures presented in Fig 1). Nevertheless, the reviewer is correct that our resolution cannot identify 1-2 kb unlabeled regions, which may remain under-replicated until G2/M. We have now addressed this point in the discussion of the revised manuscript as requested (please see page 20 lines 20-22).

Minor points

1. Figure 1b: As the authors state, the differences between the profiles are very small. In order to understand the sample distances a bit more clearly, is it possible that the authors provide a principal component (PCA) analysis for the 8 samples?

As requested by the reviewer we added a genome-wide PCA plot for all 8 samples (please see the new Fig. 1c). As can be seen in this figure the first component distinguishes between samples according to the condition (control and APH). Thus, despite a strong genome-wide correlation of RT profiles among all samples (Fig. 1b), the PCA indicates that the principle component affecting the RT is the APH treatment.

2. The FISH assays are performed on BAC clones. Is there literature on whether BAC clones form TADs and whether the TADs they form resemble their endogenous TADs?

This is an interesting question, however little is known about the ability of BAC clones within human cells to form TADs. We recently reported that BACs can be stably maintained as extra-chromosomal replicating units in human cells (Sima et al., doi: 10.1093/nar/gkx1265). RT analysis showed that BACs derived from early replicating region in the human genome maintain early replication and preferentially interacted with the endogenous early replicating regions. In contrast, BACs derived from RT transition regions or late replicating regions, replicated in mid-late S-phase and interacted with the endogenous late replicating regions. These interactions indicate sub-nuclear compartmentation, which resembles high order genome architecture (known as A/B compartments). Furthermore, as RT domains are associated with TADs (Pope et al., doi: 10.1038/nature13986), one can speculate that BACs may form a three dimensional structure when introduced into living cells.

However, in our study in order to generate FISH probes, the BAC clones were grown in bacteria, extracted and the human DNA was isolated. Moreover, this procedure includes fragmentation of the BAC DNA into 300-500 bp in length for efficient hybridization and fluorescent labelling of the DNA with fluorescent nucleotides by nick translation. The short BAC fragments are used as FISH probes on fixed cells. Hence, the human DNA that the BAC clones carry may form TADs when in the native chromosomes but not when in the form of purified fragmented and labeled probes. Similarly, we expect that the BAC probes do not affect the fixed chromosomal DNA architecture, such as TADs.

3. Please also discuss factors other than what was studied in this article that might contribute to fragility. CFS have also been linked to “origin seeding” (PMID: 21258320). This was also suggested by both Miotti et al (ORC2 density) PMID: 27436900.

As requested by the reviewer we have now expanded in the revised manuscript our discussion on the various factors contributing to chromosomal fragility. For this we added Miotto et al., PMID:

27436900 which is now ref no. 21). We elaborated in the discussion on lack of origins at CFSs as was shown by replication dynamics studies (Letessier et al, PMID: 21258320, ref 20), by tissue specific replication timing analysis (LeTsilec et al., ref 48) and by ORC2 ChIP-seq (Miotto et al., PMID: 27436900, ref 21). Please see page 3 lines 18-20 and page 20 lines 3-6 and 16-17, in the Introduction and Discussion.

4. As a final figure and to understand the contribution of the identified factors to fragility, I would suggest to include what percent of intra-TAD regions harbouring highly transcribed long genes with RT-delay are fragile sites?

We thank the reviewer for this suggestion. Analysis of intra-TAD regions harboring large highly expressed genes with RT delay shows that there are only 24 such regions, as previously described in the manuscript. Importantly, only 2 of them (8%) are located within a cytogenetically mapped fragile site (Miron et al., ref no. 40), indicating that the vast majority of these intra-TAD regions are not prone to breakage, as they might be safeguarded from the deleterious effect of replication stress. Furthermore, these results also imply that CFSs are not associated with intra-TAD regions of delayed RT within large genes, as opposed to inter-TAD regions with similar replication and transcription properties. This was now added to the manuscript (please see page 15 line 14-18 and page 18 lines 17-19).

5. Please order the figures- supplemental 8 is mentioned right after supplemental 4?

We thank the reviewer for the comment. We have now re-ordered the figures, and combined the data in previous Supplementary figures 4 and 8 which is now Supplementary Figure 9.

6. Could it be a mistake that the authors wrote they have used Bowtie to map BRU-seq reads? While I understand that BRU-seq maps nascent transcription, in a window of 30 mins (labelling), intron splicing would clearly take place. Bowtie does not map exon-exon junctions; for genes with small exons this means many reads would not be mapped. STAR (or any aligner that maps junctions) would have been appropriate choices for RNA-sequencing data. If this was

not a mistake, can the authors show that quantification of gene expression is comparable when using STAR/tophat or bowtie?

We thank the reviewer for the comment. We indeed used Bowtie to map the Bru-seq data. However, as suggested by the reviewer we re-analyzed the Bru-seq data using STAR. As described in the Methods section, the reads were first aligned to the ribosomal RNA (rRNA) repeating unit (GenBank U13369.1) and the human mitochondrial and EBV genomes using Bowtie2 (2.3.3). Reads that did not align to rRNA/chrM/EBV were mapped to hg38 using STAR (v 2.5.3a), and a STAR index created from GENCODE annotation version 27. Then RPM-normalized read densities were calculated with bedtools2 genomecov, using a scaling factor of 1000000/(number of parsed reads in library). The profiles are shown at 1kb window bins. Differential expression analysis was performed using DESeq2 (1.6.3) with betaPrior set to False. The details of this analysis were added to the revised Method section (please see pages 25-26).

It is worth noting that the STAR aligned nRNA-seq data resembled the previous Bowtie data, such that the expression of 97% of all genes, remained in the same quartile as found in the Bowtie analysis. Furthermore, no significant changes in the quartiles were found in large genes.

The analysis of STAR aligned read counts revealed no significant change in the expression of any gene, following APH treatment, please see Supplementary Fig. 7.

In summary, all the gene expression results along the manuscript and the relevant figures were updated accordingly, based on the new re-mapped data.

Reviewer #2 (Remarks to the Author):

In this manuscript Sarni et al, set to investigate the molecular mechanisms that drive common fragile sites (CFS) instability. They monitored gene expression and replication timing (RT) in the presence of aphidicolin, known to induce CFS expression, in human fibroblasts and further crossed their results with previously published Hi-C. By doing so they uncovered that CFS are very well predicted when considering the size of the gene, its expression level, its delayed RT

after replication stress and its position regarding TAD boundaries: in other words they propose that large expressed genes, that undergo delayed RT after aphidicolin treatment, and that overlap with a TAD boundary define the molecular signature of CFS.

This is an exciting study, very clearly written, presenting robust data and that is of major interest for the field. I have few suggestions that in my opinion would improve the manuscript, and a couple of recommendations for a better presentation of the results.

While the relationship between gene size, gene expression, RT delay and fragility is very well developed, I found the last part on the relationship with TAD boundaries, which is also the most novel, a little bit underdeveloped. Could the authors provide additional analyses regarding this point? (below some suggestions)

- For instance, could they show as performed in Fig. 3 and Sup Figures 4-6, the heatmap and average RT profiles, for the long genes overlapping a boundary or not, for the long expressed genes overlapping a boundary or not etc...

We thank the reviewer for the suggestion to expand and strengthen the relationship between TAD boundaries and genomic instability. We performed additional analyses as detailed below in response to all comments:

As suggested by the reviewer we have now performed additional analyses of RT profiles for large genes overlapping TAD boundaries or located within TADs. The analyses were also performed for expressed and silenced genes. The new data are presented in new Supplementary Figures 11 and 12. The analyses included:

Large genes: overall, RT profiles of large genes overlapping a TAD boundary or located within a TAD are similar to each other, showing delayed RT following APH treatment (Supplementary Fig. 11), as was found in our previous analysis for large genes in general.

Silenced large genes: Further analysis of RT for silent large genes, showed that both TAD boundary overlapping genes and genes located within a TAD domain are late replicating with no significant RT delay following APH (Supplementary Fig. 12a,c,e).

Expressed large genes: RT analysis of expressed large genes revealed that they are delayed following APH treatment, regardless of their 3D architecture (overlapping a TAD boundary or not). Please see Supplementary Fig. 12b, d and f.

Of note is that although the RT profiles of inter- and intra-TAD large genes are similar, there is a significant delay in the RT of expressed large genes spanning a TAD boundary compared to expressed large genes in intra-TADs. Please see Supplementary Fig. 12f.

Altogether, these new analyses indicate that the TAD boundary overlapping large expressed genes are significantly more susceptible to replication delay under stress conditions than large expressed genes located within TADs. Please see the changes in the text explaining the new results from page 15 line 18 to page 16 line 7.

- Could they draw replication timing around all TADs boundaries? Around TAD boundaries located in large active genes ? etc...

As requested by the reviewer we have now performed additional analysis of the RT profiles around all TAD boundaries. The results show that the average RT of TAD boundaries is not affected by APH treatment (New Supplementary Fig. 13a), implying that the TAD boundary per se does not affect the RT program.

Next we analyzed the RT at TAD boundaries within genes and found a RT delay only in TAD boundaries within large genes (Supplementary Fig. 13b). We further explored the RT of TAD boundaries in silent compared to expressed large genes (Supplementary Fig. 13c,d). The results show that the RT of TAD boundaries within silent large genes is late replicating and is not affected by APH. In contrast, the RT of TAD boundaries located within expressed large genes is early replicating and is delayed by APH treatment. Moreover, TAD boundaries are positioned at the most delayed regions within these genes. The results raise the possibility that TAD boundaries within delayed expressed large genes may perturb the completion of DNA replication in these regions, implying that the 3D architecture affects the RT of large expressed genes under replication stress. Please see the changes in the manuscript describing the new results on page 16 lines 8-22.

- One would expect more CTCF binding at these CFS. Using already published dataset for CTCF distribution, could the authors show the CTCF enrichment on expressed large genes that are fragile, compared to expressed large gene that are not fragile? And on large/ expressed/RT-delayed/ fragile versus large/ expressed/RT-delayed/ non-fragile?.

TAD boundaries were indeed found to be enriched with CTCF binding (for example by Dixon et al., doi: 10.1038/nature11082; Nora et al., doi: 10.1038/nature11049). However, analysis of CTCF binding across the genome, revealed that the vast majority of CTCF binding is at non-TAD boundary regions (Dixon et al., doi: 10.1038/nature11082; Nora et al., doi: 10.1038/nature11049). As requested by the reviewer we analyzed publically available CTCF ChIP-seq data of HFF cells from the ENCODE (GEO:GSM1022644). Our analyses summarized in Supplementary Fig 14, are in agreement with the previous reports and show that TAD boundaries are enriched with CTCF binding compared to randomly selected size matching loci across the genome (Supplementary Fig. 14a) and to their flanking regions (Supplementary Fig. b). Importantly however, CTCF binds genome-wide, and thus the vast majority of binding sites are not associated with TAD boundaries (Supplementary Fig. 14c). We then analyzed CTCF binding at large genes overlapping a TAD boundary and found that 423/442 are bound by CTCF, as expected. However, the majority of large genes located within a single TAD domain are also bound by CTCF, as 182/196 of them were bound by CTCF. Thus, CTCF probably does not play a significant role in fragility as most large genes are bound by CTCF, whether they are intra-TAD or spanning a TAD boundary. Please see page 17 lines 11-22.

Another quantitative value that might be useful is the insulation score, which helps to define TAD boundaries.

We would like to clarify that the “TAD separation score” we used in this manuscript is actually insulation score, as described in the HiCEplorer manual, under the hiCFindTADs function (<https://hicexplorer.readthedocs.io/en/latest/content/tools/hicFindTADs.html#hicfindtads>):

“TAD boundaries locations are stored in the boundaries files, domains.bed file contains the TAD locations, score files contain TAD separation score, or the so-called TAD insulation score, in various formats.”

- Fig3d, shows that CFS can also take place on large, silent, genes, that do not display changes in RT: does this specific category overlap TAD boundaries?, can the authors show the average CTCF enrichment compared to other large, silent, not fragile genes?

We thank the reviewer for this comment highlighting the need for a more clear description of the analysis in Fig 3d. We wish to emphasize that CFSs were mapped using the low resolution G-banding technique. These cytogenetic bands may harbor several large genes. For example within the cytogenetic band chr2q22 (presented in Fig. 2a-d) there are 5 large genes, however the FISH mapping showed that the fragility in this chromosomal band is in a region harboring only one of the large genes (*GTDC1*), which is expressed and delayed, while the other four large genes are not part of the fragility region (as detailed in pages 13 and 14 of the manuscript). Hence, not all large genes within the cytogenetically mapped CFS bands are part of the core fragility region. Importantly, silent large genes were found to be non-fragile (Fig 2). We have now clarified along the text that a cytogenetically mapped CFS refers to a cytogenetically band in which only part is fragile. See for example page 13 lines 21-23.

Regarding the TAD boundary overlapping large silenced genes within cytogenetically mapped CFSs. Since these genes do not have the entire fragility signature (they are silent and are not delayed) these genes are not within the core fragility region (Fig 2), even if they span a TAD boundary.

Minor points

- In Fig. 3, Fig S4, Fig. S6 the legend for gene size goes to red (dark pink?) for very large gene to red for very small genes: I would recommend changing the color code to really see differences between large and small

We thank to reviewer for the helpful comment, and as suggested we changed the color code in all the relevant figures in the revised manuscript.

- For all the heatmaps please make sure that the scale for RT timing data between Aphidicolin treated and control cells is the same.

We thank the reviewer also for this helpful comment, and as suggested we matched the RT profile scales of control and APH treated cells in the relevant figures in the revised manuscript.

- I would recommend changing the term “inter-TAD” as this is very confusing. Maybe something like “overlapping TAD boundary”, or “TAD boundary containing genes”

As suggested by the reviewer we have changed the term “inter-TAD” to “overlapping TAD boundary” along the manuscript.

- Typo in the abstract first line: “Common fragile sties”, please change

We apologize for this typo which was fixed.

- For the FISH experiment, can you indicate how many events were scored? I would also recommend explaining a little bit more Fig2c-d (on p8)

The numbers of chromosomes analyzed are now described in the legend of Figure 2.

In addition, as requested by the reviewer, we elaborated the explanation of the FISH experiments presented in Figure 2c,d (please see page 9 lines 1-5).

- Typo on Fig. S7 in the legend “No chnage”

We apologize for this typo.

However, the figure was changed according to a suggestion made by reviewer 1 (minor comment no. 6). We have remapped the Bru-seq reads and re-performed differential gene expression analysis using DESeq2 based on the new raw counts. The analysis showed no significant change in gene expression following APH treatment. Hence, Supplementary Figure 7 was updated and the legend was removed.

REVIEWERS' COMMENTS:

Reviewer #1 (Remarks to the Author):

I am very satisfied with the author's answers to all my concerns. Their work was thorough and exemplary. I have no further comments.

Looking forward to its publication.

Ildem Akerman

Reviewer #2 (Remarks to the Author):

In their revised manuscript, Sarni et al adequately addressed all my previous comments. The extended analysis for the relationship between TAD boundaries and fragility significantly improved the manuscript. I just have a little comment that is unclear to me: Fig S2B, the RT signature 4 shows delayed RT upon APH but seems to go from late replication to even later, while on Fig 2e and 2f it seems that Signature 4 goes from early replication to less early replication (but, if I understood correctly it should still show a log RT ratio above 0). Can the author explain and comment on that?

P15 the following sentence should read: "Importantly, only 2/24 (8%) cytogenetically mapped CFSs harbor a RT-delayed 322 transcribed large gene located within a TAD domain" (replace with by within)

Gaelle Legube

Point by point response to the reviewers' comments on the revised manuscript

June 7, 2020

We have addressed the comments of the reviewers. Please see below a detailed point by point response with our remarks in blue.

Reviewer #1 (Remarks to the Author):

I am very satisfied with the author's answers to all my concerns. Their work was thorough and exemplary. I have no further comments.

Looking forward to its publication.

Ildem Akerman

We thank the reviewer for the valuable and constructive comments on the original version of the manuscript which contributed to the manuscript and improved it.

Reviewer #2 (Remarks to the Author):

In their revised manuscript, Sarni et al adequately addressed all my previous comments. The extended analysis for the relationship between TAD boundaries and fragility significantly improved the manuscript. I just have a little comment that is unclear to me: Fig S2B, the RT signature 4 shows delayed RT upon APH but seems to go from late replication to even later, while on Fig 2e and 2f it seems that Signature 4 goes from early replication to less early replication (but, if I understood correctly it should still show a log RT ratio above 0). Can the author explain and comment on that?

We thank the reviewer for the valuable and constructive comments on the original version of the manuscript which contributed to the manuscript and improved it. Additionally, we thank the reviewer for drawing our attention to our unfortunate mistake in Figure S2b.

We mistakenly used to generate this figure the wrong RT data set. We apologize for this. We have now went back to the original RT data and re-generated the figure. As can be seen in the corrected figure S2b the RT signature 4 shows delayed RT upon APH treatment from early to mid S-phase, as expected from the signatures shown in Figure 1d and Figure 1e. In addition, all RT signatures are significantly differentially replicated.

P15 the following sentence should read: "Importantly, only 2/24 (8%) cytogenetically mapped CFSs harbor a RT-delayed 322, transcribed large gene located within a TAD domain" (replace with by within)

Gaëlle Legube

We thank the reviewer for the comment, "with" was corrected to "within".